# Current Mode Control of a Series LC Converter Supporting Constant Current, Constant Voltage (CCCV)

**Michael Heidinger \*** , **Qihao Xia, Christoph Simon, Fabian Denk, Santiago Eizaguirre, Rainer Kling and Wolfgang Heering**

Karlsruhe Institute for Technology (KIT)—Light Technology Institute, Engesser Straße 13, 76131 Karlsruhe, Germany
**\*** Correspondence: michael.heidinger@kit.edu; Tel.: +49-721-608-47852

**Abstract:** This paper introduces a control algorithm for soft-switching series LC converters. The conventional voltage-to-voltage controller is split into a master and a slave controller. The master controller implements constant current, constant voltage (CCCV) control, required for demanding applications, for example, lithium battery charging or laboratory power supplies. It defines the set-current for the open-loop current slave controller, which generates the pulse width modulation (PWM) parameters. The power supply achieves fast large-signal responses, e.g., from 5 V to 24 V, where 95% of the target value is reached in less than 400 µs. The design is evaluated extensively in simulation and on a prototype. A match between simulation and measurement is achieved.

**Keywords:** control; current mode control; voltage control; transfer function; power converter; soft-switching converter; battery charging

---

## 1. Introduction

By the use of soft-switching converters, highly efficient DC/DC converters can be built. One possible topology, a series LC (SLC) converter is shown in Figure 1. The topology is similar to a series resonant converter, but it operates in a non-resonant push–pull mode [1]. In contrast to a dual active half-bridge converter, the two secondary side active output switches are replaced with diodes [2].

A detailed time domain analysis for calculating the SLC output current, operated above the LC resonance frequency, was published recently [1]. Current literature proposes a voltage-to-voltage transfer function [3,4]. We split the voltage-to-voltage converter in a cascaded structure [2,5] for enhanced performance. A master controller sets the SLC output current, while a slave open-loop transfer function controls the switching period, duty cycle, and pulse-skipping. By the use of current mode control, one pole is eliminated in the control loop [6]. The master voltage controller supports constant current, constant voltage (CCCV) operation, which is required, e.g., for battery charging [7] or laboratory power supplies.

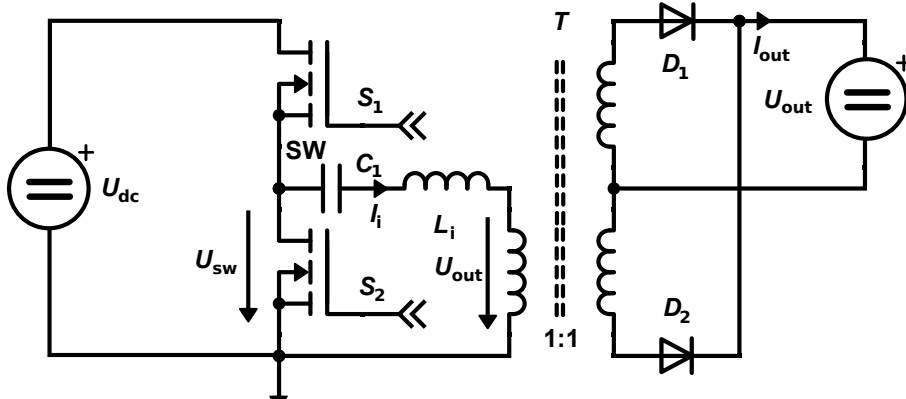

**Figure 1.** The schematic of the series LC converter is identical to the series resonant converter. However, the resonant capacitor $C_1$ is chosen large and acts as a DC blocking capacitor. The converter is operated far above its resonance frequency.

## 2. State-of-the-art

State-of-the-art soft-switching converters, e.g., series resonant converters (SRC) or LLCs, are modeled using a voltage transfer function [3,4]. This function can be derived, for example, by the first harmonic approximation. Current research modeled resonant converters operated far above the resonant frequency, so-called series LC converters, by a voltage-to-current transfer function [1]. Hence, the idea of current mode control suggests itself.

Current mode control has been used in flyback converters for a very long time [8]. Thereby, the loop is split into a master voltage controller and a slave current mode controller [5,8]. This approach has already been shown for dual active half-bridge [2] and resonant converters [6]. Thereby, current mode control has a multitude of advantages [6]. State-of-the-art closed loop current mode control uses a low pass filter [6], resulting in a reduced bandwidth. This work uses the open-loop voltage-to-current transfer function, which has a higher bandwidth, to further enhance performance [1].

Series resonant converters are typically controlled by switching frequency only [4]. However, pulse skipping and duty cycle modulation have also been presented for current limiting [9]. Research also demonstrated linear open-loop feed-forward control [10], whereas this work uses non-linear open-loop feed-forward control, based on [1].

## 3. Fundamentals

Equation (1) formulates the SLC converter output current $I_{cc}$ as a function of the input voltage $U_{dc}$ and output voltage $U_{out}$ based on the control parameters duty cycle $D$ and switching period $t_p$ [1]. By adding pulse skipping, where $p_o$ and $p_c$ represent the pulse skipping parameter defined in Figure 2, a very large output current range is achieved.

$$I_{cc} = \frac{p_o}{p_c} \frac{D(1-D)U_{dc}^2 - U_{out}^2}{4L_i U_{dc}} t_p \tag{1}$$

As the input voltage and also the output voltage are monitored in (1), this equation allows a very high rejection ratio [1]. The output voltage is $U_{out}$, and the measured output voltage is referred to as $U_{meas}$. As a large input voltage ripple can be rejected, this allows for a reduction of the DC link capacitance $C_{dc}$. This enables the use of film capacitors instead of electrolytic capacitors, extending the estimated service life of the power supply.

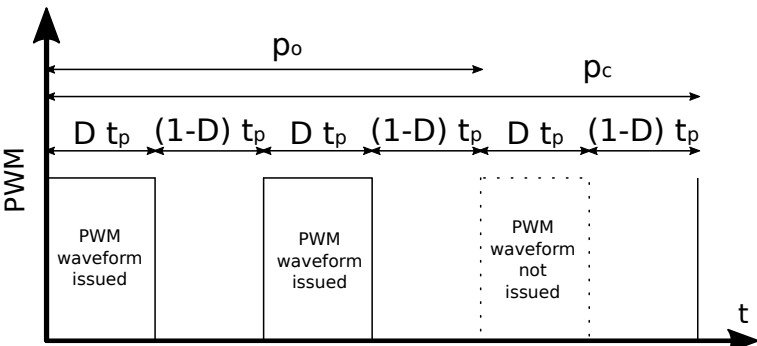

**Figure 2.** A PWM pulse skipping waveform is shown, where the period ($t_\mathrm{p}$), duty cycle ($D$), and pulse skipping ontime ($p_\mathrm{o} = 2$) and pulse skipping period ($p_\mathrm{c} = 3$) are highlighted.

The proposed control diagram is shown in Figure 3. The control is based on four elements: {1} The master voltage controller sets the current to the slave current controller. {2} the slave current mode controller is an open-loop control transfer function based on (1). The current controller sets four parameters to the pulse width modulation (PWM) modulator {3}. The PWM modulator generates the PWM output waveform for the series LC converter {4}.

The controller is implemented on a digital signal processor (DSP), as (1) requires non-linear calculations. ADCs digitize the input voltage, output voltage, and output current for the CCCV control. The DSP integrates a PWM module, generating the gate signals for the half-bridge.

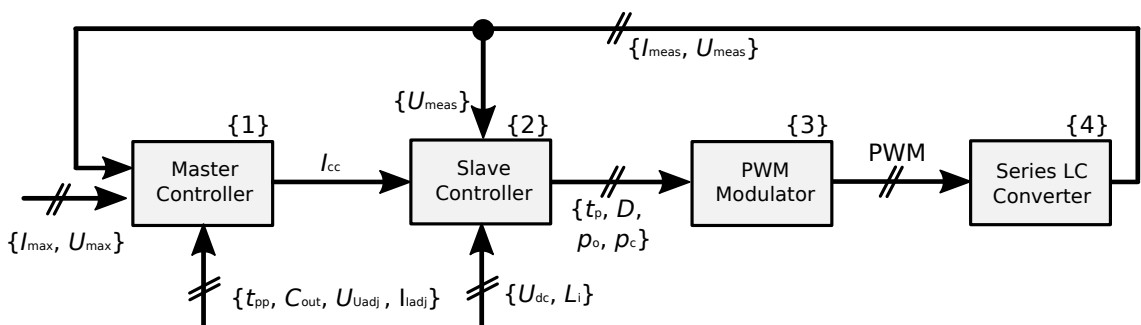

**Figure 3.** Proposed control diagram for the series LC converter. The converter is split into a master controller, implementing constant current, constant voltage (CCCV) control, and an open-loop slave controller. The MCUs pulse width modulator (PWM) is used to generate the SRCs gate signals.

## 4. Master Voltage Mode Controller

The master voltage mode controller is shown in detail in Figure 4. It consists out of two controllers: one to control the voltage and one to control the current. Both operate in parallel, and the minimum value is selected to control the set current $I_\mathrm{cc}$ for the constant current controller.

The sensed current $I_\mathrm{sense}$ should be filtered to prevent systems oscillation when a capacitive load is connected. For the prototype, presented in Section 7, a second order low pass filter with a cutoff frequency of 16 kHz is used. Both controllers are discussed in detail in the following subsections.

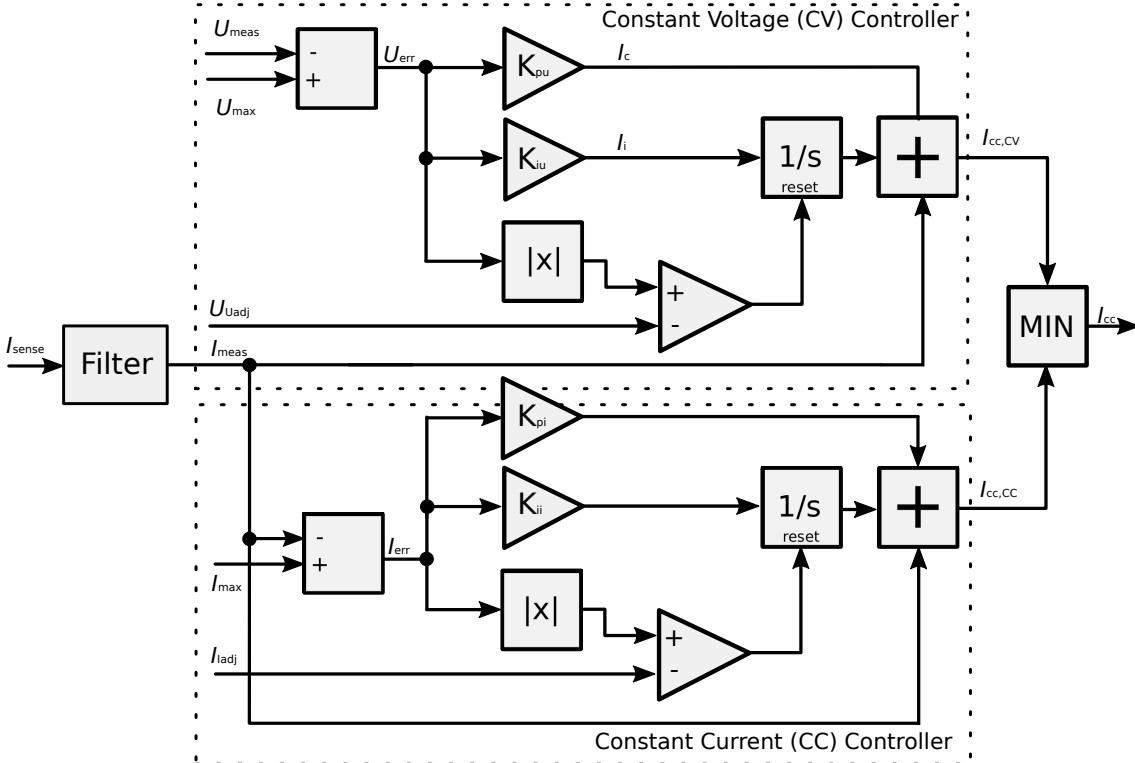

**Figure 4.** The master controller implements the CCCV functionality. The current and the voltage controllers operate in parallel, and the smaller set value is selected for the set current $I_{cc}$.

### 4.1. Constant Voltage Controller

The constant voltage controller limits the maximal output voltage to $U_{max}$. We design the voltage control loop in Figure 4 based on circuit analysis of the output capacitor $C_{out}$. The required set current $I_{cc,CV}$ is expressed in (2). The filtered current is designated $I_{meas}$.

$$I_{cc,CV} = I_{meas} + I_c + I_i \tag{2}$$

The equalization current ($I_c + I_i$) is calculated by a PI regulator. The proportional gain is chosen on the charge balance observation: we calculate the proportional equalizing current $I_c$ as a function of the output charge.

$$Q = I_c \cdot t_{pp} = C_{out}(U_{max} - U_{meas}) \tag{3}$$

$$I_c = \frac{C_{out}(U_{max} - U_{meas})}{t_{pp}} \tag{4}$$

The time constant $t_{pp}$ in (5) is chosen with respect to the maximal digital regulator control loop period. Stability was observed by using a factor of 1/4 or less at a control loop frequency of 85.750 kHz. Hence, the P regulator gain can be formulated as:

$$K_{pu} \leq \frac{C_{out}}{4 \cdot t_{pp}} \tag{5}$$

The DC voltage accuracy is enhanced by increasing the proportional gain $K_{pu}$. Referring to (5), the output capacitance is proportional to the maximal proportional gain.

As shown in Figure 4, an I regulator with reset is used to achieve stationary accuracy of the control. It adjusts a typically small error. To reduce overshoot, it is only activated if the error is lower than an absolute minimal error. We name this minimal voltage $U_{Uadj}$.

*4.2. Constant Current Controller*

The constant current controller limits the output current to $I_{max}$. Previous research already demonstrated that the slave output current accuracy $I_{cc}$ is better than 7% [1]. Therefore, the output current is directly forwarded to the limiter. To compensate for inaccuracies, an additional PI regulator is used. If the absolute error is larger than $I_{Iadj}$, the I regulator is reset to reduce overshoot for large signal responses.

*4.3. Acoustic Noise*

This design uses multilayer ceramic capacitors (MLCCs) as output capacitors. They may emit acoustic noise due to the capacitors piezoelectric dielectric. If the master P gain is chosen close to the critical gain, noise is emitted. The acoustic noise is reduced by lowering the P gain or choosing low noise MLCCs.

Our experiments concluded that the following control loop gain eliminated the noise at the cost of a slightly slower step response:

$$K_{pu} \leq \frac{C_{out}}{9 \cdot t_{pp}}. \tag{6}$$

## 5. Slave Current Controller

The slave current controller receives the set current $I_{cc}$ from the master controller. It is responsible for selecting the appropriate modulation scheme; it chooses between adjusting the switching period, duty cycle, or pulse skipping, as shown in Figure 5.

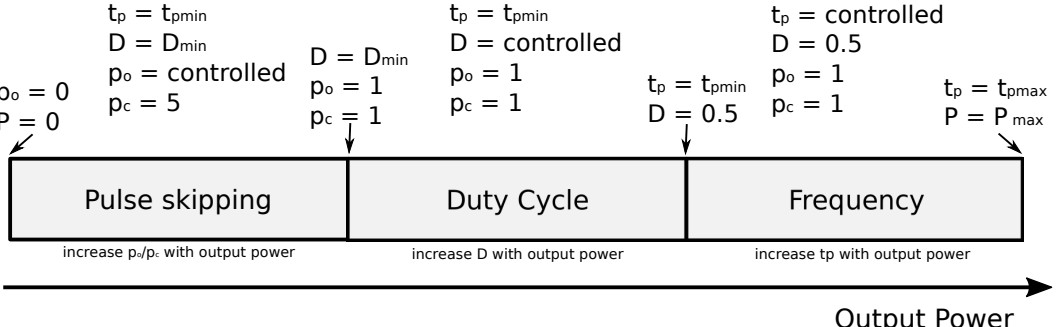

**Figure 5.** The slave current mode controller controls the frequency ($f_p = 1/t_p$) for high output power, the duty cycle ($D$) for medium output power, and uses pulse skipping ($p_o/p_c$) for low output power. Changing the modulation scheme maximizes the output current range.

For adjusting the output power, we propose the modulation strategy shown in Figure 5. Pulse skipping is used for the lowest possible output power. It is identified by missing PWM pulses. The number of pulses may range between zero and $p_c$. At medium output power, the duty cycle is in the range of $D = D_{min,abs}$ to $D = 0.5$, while the minimal switching period is used. This modulation is referred to as duty cycle modulation. The minimal duty cycle is a design parameter and is chosen to $D_{min,abs} = 0.2$.

At very high output power, the switching period is increased until the maximal allowed $t_{p,max}$ is reached. Frequency modulation uses a duty cycle of 0.5 and a variable switching frequency. In the case where $D_{max} < 0.5$, duty cycle ramp-up is used. It uses the minimal switching frequency while increasing $D_{max}$ by steps of $\Delta D$. Duty cycle ramp-up reduces stress on capacitor $C_1$ and prevents overcurrent.

The implementation of the algorithm for determining the appropriate modulation scheme is visualized in Figure 6 and is discussed in detail next.

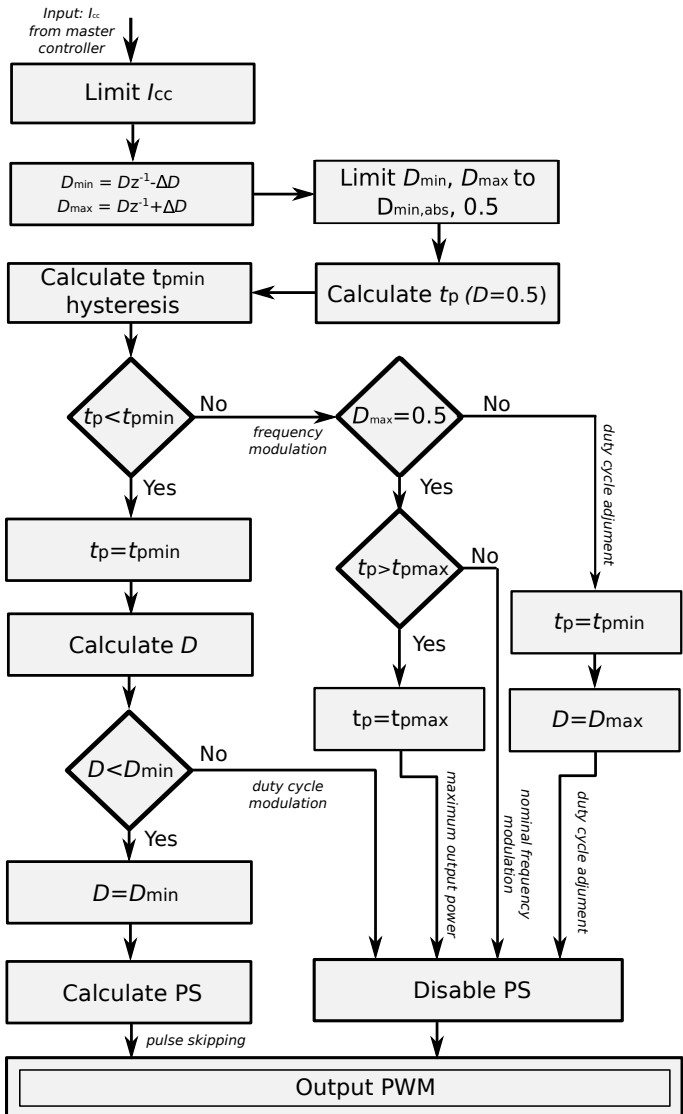

**Figure 6.** Slave current mode controller algorithm.

## 5.1. Initial Calculus

The algorithm shown in Figure 6 first limits $I_{cc}$ to a positive value. Second, the minimal and maximal duty cycles are calculated, which are next limited. Third, the period $t_P$ is determined using (7), which is based on (1), using a duty cycle of $D = 0.5$.

$$t_P = \frac{16 L_i U_{dc} I_{cc}}{U_{dc}^2 - 4 U_{meas}^2}. \tag{7}$$

Next, the minimum period $t_{p,min}$ is calculated, which is a constant value, with an additional hysteresis. In the simulation and experiments, no hysteresis was utilized. The minimum switching frequency $t_{p,min}$ is typically chosen in such a manner that soft-switching of the half-bridge is still achieved.

## 5.2. Frequency Modulation

If the period $t_P$ is larger than $t_{p,min}$, switching frequency modulation is used. To prevent unintentional false-triggering due to capacitor $C_1$ displacement current, the minimal period $t_{p,min}$ is used during duty cycle adjustment. A duty cycle ramp-up is in progress when $D_{max} < 0.5$.

The maximum switching period is chosen based on (8). The constant $k$ should be chosen in the range of 0.5 to 0.7, depending on the design goals. A lower $k$ results in less slave controller error, while a higher $k$ allows a higher output current.

$$t_{p,max} = k\pi \sqrt{L_i C_1} \tag{8}$$

### 5.3. Duty Cycle Modulation

If the period $t_p$ is smaller than $t_{p,min}$, duty cycle modulation is used. The duty cycle is determined by (1). As a quadratic equation has two results, one has to be chosen. To limit the voltage stress on $C_1$, the smaller result is used. This results in the following equation:

$$D = \frac{U_{dc}\, t_{p,min} - \sqrt{\left(U_{dc}^2 - 4U_{meas}^2\right) t_{p,min}^2 - 16 I_{cc} L_i U_{dc}\, t_{p,min}}}{2U_{dc}\, t_{p,min}}. \tag{9}$$

### 5.4. Pulse Skipping

If the calculated duty cycle is smaller than the minimum duty cycle $D_{min}$, pulse skipping is used. The pulse modulation is calculated according to the following formula, which is converted to the number of on-pulses ($p_o$) and total number of periods ($p_c$). An example for a PWM waveform with pulse skipping is given in Figure 2.

$$I_{cc} = \frac{p_o}{p_c} \frac{D_{min}(1 - D_{min})U_{dc}^2 - U_{meas}^2}{4L_i U_{dc}} t_p \tag{10}$$

Currently, a fixed pulse skipping period $p_c$ is used. However, the Farey method could also be used to determine a more accurate ratio [11]. If $p_o < 0.5$, the PWM output is disabled. Thereby, very low pulse counts are achieved. To prevent acoustic noise by pulse skipping, the pulse skipping frequency should be larger than the maximal audible frequency $f_a = 20\,\text{kHz}$:

$$\frac{1}{p_c\, t_{p,min}} > f_a. \tag{11}$$

Pulse skipping introduces a significant amount of output ripple. Currently, output ripple can be reduced by increasing the output capacitor $C_{out}$. To further reduce output ripple, an appropriate LC filter and its impact on the output response could be investigated.

### 5.5. Voltage Stress on $C_1$

The voltage $U_c$ on the offset capacitor $C_1$ is calculated using the following equation [1]:

$$U_c = D U_{dc}. \tag{12}$$

To limit the voltage slope stress on the offset capacitor $C_1$ and slow down its aging, the duty cycle is only changed slowly. Currently, a value of $\Delta D = 0.02$ per iteration is used to limit the stress on the DC blocking capacitor $C_1$.

### 5.6. Input Voltage Range

The minimal input voltage must be at least twice the output voltage when a duty cycle of 50% is applied. Furthermore, a transformer ratio of 1:1 and no output current is assumed. The following equation is derived based on (1).

$$U_{out,max} = 0.5\, U_{dc} \tag{13}$$

To determine the minimum DC link voltage for a given output voltage $U_{out}$ and an output current $I_{out}$, (1) is solved for $U_{dc,min}$.

$$U_{dc,min} = \frac{2\sqrt{U_{out}^2 t_{p,max}^2 + 16 I_{cc}^2 L_i^2} + 8 I_{cc} L_i}{t_{p,max}} \tag{14}$$

The minimum input voltage calculated by (14) is plotted for the converter as a function of output voltage in Figure 7. It shows that, the SLC converter allows a lower minimum input voltage when a reduced output voltage range is sufficient. Therefore, input and output voltage range may be traded off.

Equation (14) shows that the larger the maximum switching period $t_{c,max}$ is, the less influence the output current $I_{out}$ has on the minimal input voltage $U_{dc,min}$. This is visualized in Figure 8. The minimal input voltage $U_{dc,min}$ is shown as a function of the maximal cycle time $t_{p,max}$ at an output voltage of $U_{out} = 25$ V. According to (8), the output current range is increased by choosing a larger capacitance $C_1$.

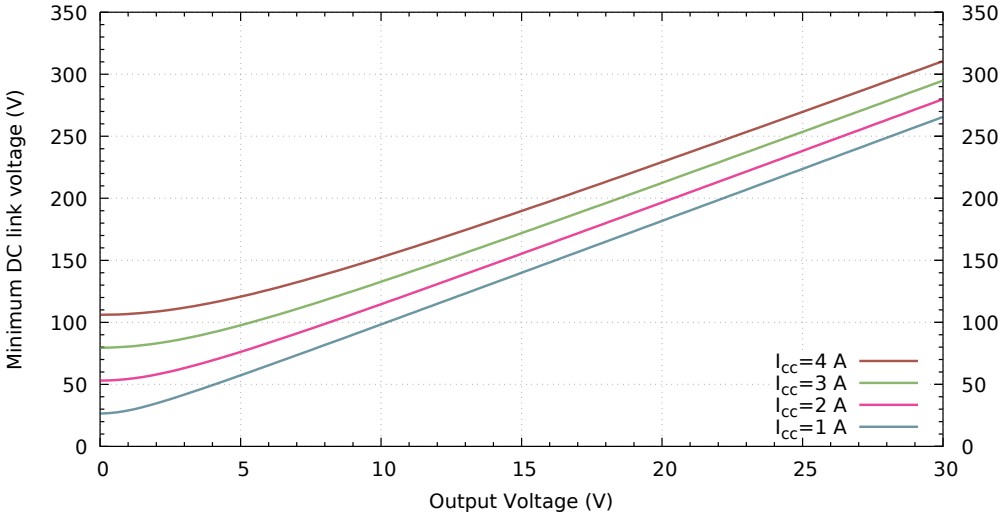

**Figure 7.** The minimum DC link voltage is plotted over the output voltage of the converter. The output current $I_{cc}$ is shown as a parameter.

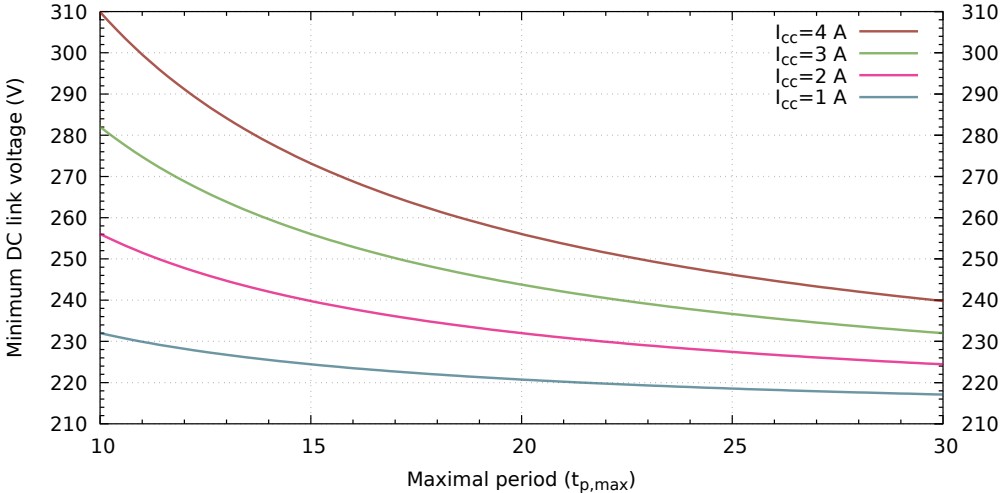

**Figure 8.** The minimum DC link voltage is plotted over the maximal allowable period for an output voltage $U_{out} = 25$ V. The output current $I_{cc}$ is shown as a parameter.

## 6. Modulator

The DSP PWM unit generates the gate signals. It has four input parameters: The period $t_p$, the duty cycle $D$, the number of emitted pulses $p_o$, and the number of pulses per period $p_c$. An example is depicted in Figure 2. The period $t_p = \frac{1}{f_{sw}}$ is the inverse of the switching frequency, and the duty cycle states the ratio of the PWM high period. A switching cycle can be skipped by pulse skipping. The pulse skipping ontime $p_o$ states how many PWM pulses are emitted during a pulse skipping period $p_c$.

## 7. Simulation and Experimental Results

The following section covers the simulation and measurement results for the CCCV converter.

### 7.1. Measurement Setup

To verify operation, the circuit was simulated with the software PLECS and tested in an experimental setup. The build converter prototype is shown in Figure 9. The test parameters are shown in Table 1, unless otherwise noted in the measurement description. For the simulations and experiments, a load resistor of $R_{load} = 10\,\Omega$ was connected to the output.

Four experiments were carried out on the prototype: {1} a constant voltage step response test, {2} a constant current step response test, {3} a load response test, and {4}, the CCCV step response. For each test setup, the corresponding output current and voltage were measured. In addition to each experiment, output voltage and current were simulated as well. The depicted duty cycle and switching period are extracted from simulation only. In the fifth simulation, the converter was operated at 230 V, 50 Hz AC, demonstrating its ability to reject a large input voltage DC link ripple.

**Table 1.** Test setup parameters and conditions.

| Element/Parameter | Value |
|:---:|:---:|
| $U_{in}$ | 325 V |
| $T_{ratio}$ | 4.2:1 |
| $L_i$ | 110 μH |
| $C_1$ | 470 nF |
| $C_{out}$ | 110 μF |
| $P_{out,max}$ | 62.5 W |
| $K_{pu}$ | 1.0 |
| $K_{iu}$ | 857.5 |
| $U_{Uadj}$ | 0.05 $U_{set}$ |
| $K_{pi}$ | 20 |
| $K_{ii}$ | 17,150 |
| $I_{Iadj}$ | 0.05 $I_{set}$ |
| $\Delta D$ | 0.02 |
| $D_{min,abs}$ | 0.2 |
| $t_{p,min}$ | 5 μs |
| $t_{p,max}$ | 15.8 μs |
| $k$ | 0.7 |
| $p_c$ | 5 |
| $f_{control}$ | 85.750 kHz |

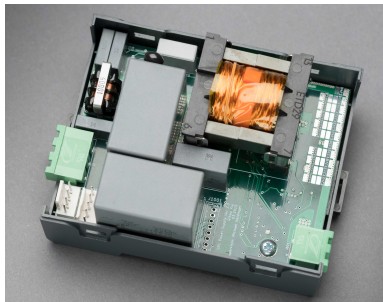

**Figure 9.** The prototype is mounted in a DIN rail case measuring 85 mm by 65 mm. It avoids electrolytic capacitors and replaces them with film capacitors to achieve a longer service life.

## 7.2. Voltage Step Response

The constant voltage controller limits the allowable output voltage. To verify the constant voltage controller, the maximal output voltage $U_{max}$ was increased in a step response at $t = 0$ from 5 V to 24 V; the step response is shown in Figure 10.

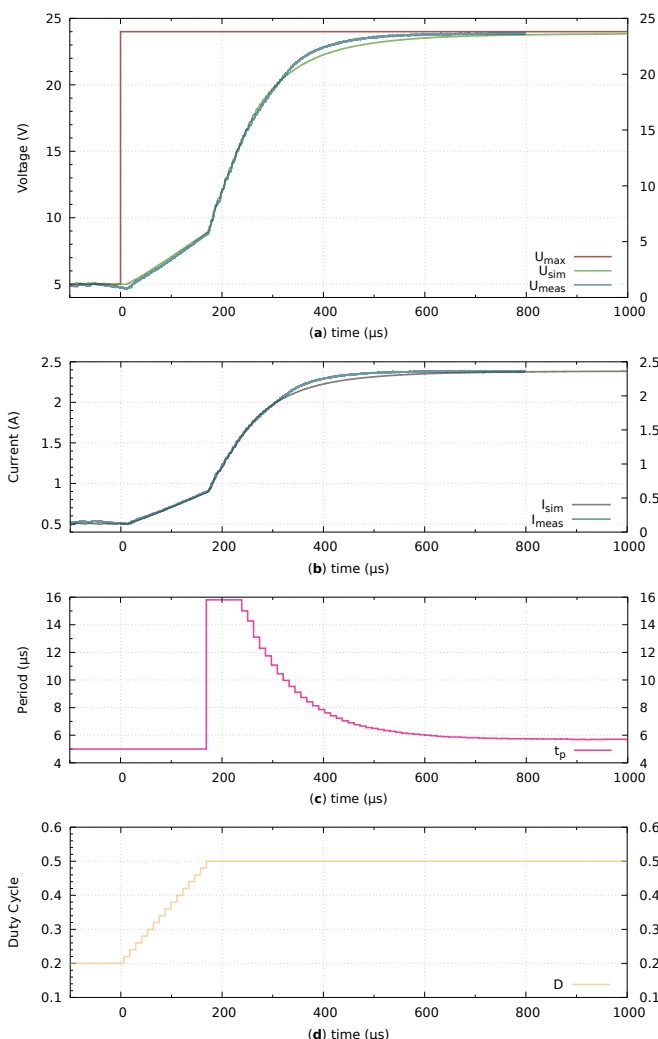

**Figure 10.** The constant voltage operation of the control was verified: the output voltage limit $U_{max}$ was increased in a step response from 5 V to 24 V (**a**) at $t = 0$. The simulated and measured output voltages are shown in (**a**). The simulated and measured output current are shown in (**b**). The simulated switching period $t_p$ is shown in (**c**), while the simulated duty cycle $D$ is shown in (**d**).

Before the step at $t < 0$, the duty cycle $D$ was limited to $D_{min} = 0.2$ as pulse skipping was used. When the output power increased after $t = 0$, the duty cycle was increased during duty cycle ramp-up. The duty cycle $D$ increased each control loop iteration by steps of $\Delta D = 0.02$ to $D = 0.5$. At $t \approx 180$ µs, the switching period was limited in frequency modulation to provide the maximal save output current while ensuring over-resonant operation. When the output voltage $U_{meas}$ was about to reach the maximal voltage $U_{max}$, the switching frequency was reduced. The converter remained in frequency modulation.

In Figure 10, the output voltage rose fast and reached 95% of the target output voltage in less than 400 µs. A transition from pulse skipping modulation at low load to switching frequency modulation at high load was demonstrated.

When the experiment's output voltage $U_{meas}$ is compared to the simulated output voltage $U_{sim}$ in Figure 10, a match is observed. The non-congruence between $t \approx 300$ µs and $t \approx 600$ µs arises due to the non-linear MLCC output capacitance.

## 7.3. Current Step Response

The constant current controller limits the maximum allowable output current. For verification, the maximal output current $I_{max}$ was increased from 1 A to 2 A. The output current step response is shown in Figure 11. Before $t = 0$, the converter operated in pulse skipping mode. Pulse skipping generated a significant ripple on the output voltage, as the effective switching frequency is very low.

At $t = 0$, the output current $I_{max}$ was increased from 1 A to 2 A, and the duty cycle was increased during duty cycle ramp-up from $D = 0.2$ to $D = 0.5$ in increments of $\Delta D = 0.02$ per control iteration, while using the minimal period $t_p$ to prevent overcurrent triggering. To charge the output capacitor $C_{out}$ fast, the slave controller first utilized frequency modulation, but switched back to duty cycle modulation at $t \approx 230$ µs as the output capacitor is almost charged.

Current control reached 95% of the output current target in $t \approx 300$ µs. At $t < 400$ µs, the fine adjustment by the PI regulator was complete. The converter did not overshoot on the output current.

When the measured output current from the experiment $I_{meas}$ is compared to the simulated output current $I_{sim}$ in Figure 11, a match is observed. The slight difference is due to the non-linear MLCC output capacitance.

## 7.4. Load Response

The load response monitors the output voltage change while the load is increased. For this experiment, shown in Figure 12, the output voltage was held constant at $U_{max} = 5$ V, while an external current load was increased from $I_{meas} = 0.5$ A to $I_{meas} = 2$ A at $t = 0$. The output capacitor was chosen for this experiment to $C_{out} = 160$ µF. Based on other system requirements, the control loop speed was reduced to 75 kHz for this experiment. To completely eliminate overshoots, the voltage PI controller was modified to $K_{pu} = 0.9$ and $K_{iu} = 343$. For reference, the simulated standard parameter response is shown in orange. It shows a simulated overshoot of less than 1%.

Before $t < 0$, the converter operated in pulse skipping mode, explaining the significant output ripple. At $t = 0$, the output current is $I_{out}$ increased to 4 A. Therefore, the slave controller utilized duty cycle ramp-up, in which it increased the duty cycle by $\Delta D = 0.02$ per control cycle to $D = 0.5$. This limits capacitor stress on $C_1$ and prevents overcurrent. Because of that, the converter could not supply sufficient output power, hence the output voltage decreased. When the duty cycle adjustment was completed, at $t \approx 200$ µs, the controller switched to frequency modulation at $D = 0.5$. No overshoot of $U_{meas}$ was observed. The controller then remained in frequency modulation. The simulation matches the measurement. It must be noted, that the presented load jump represents the worst case.

The load response may be improved by not using the minimal switching frequency during duty cycle adjustment. However, in that case, the overcurrent detection level has to be increased.

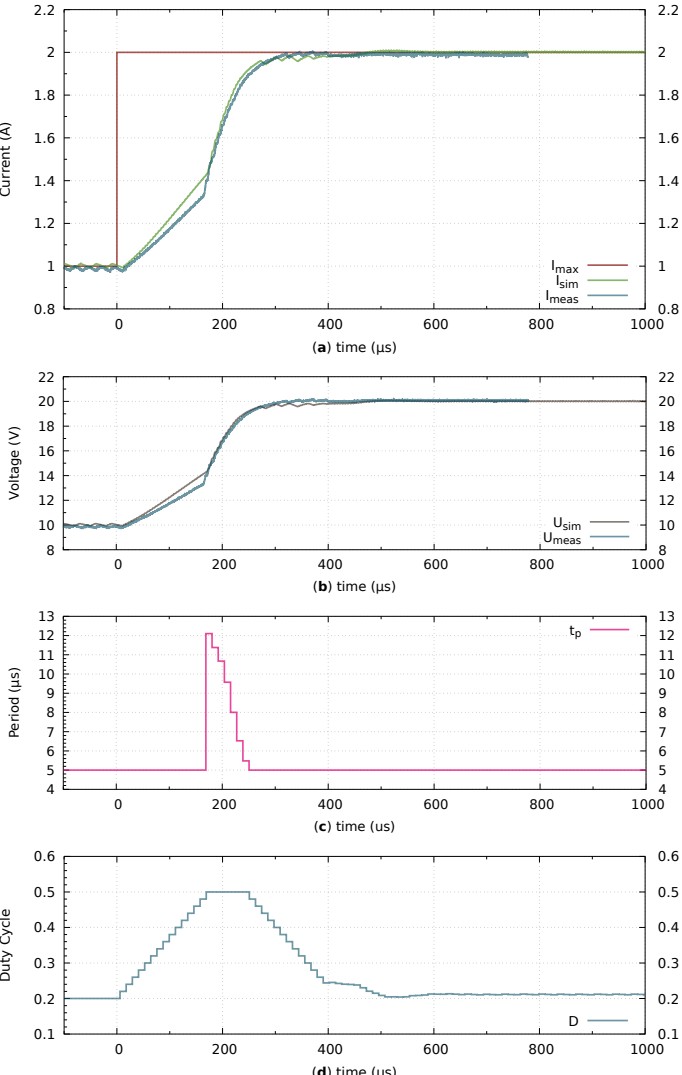

**Figure 11.** The constant current operation of the control was verified: the maximum output current $I_{max}$ was increased in a step response from 1 A to 2 A (**a**) at $t = 0$. The simulated and measured output currents is shown also in (**a**). The simulated and measured output voltage is shown in (**b**), while the simulated switching period $t_p$ is shown in (**c**), and the simulated duty cycle is shown in (**d**).

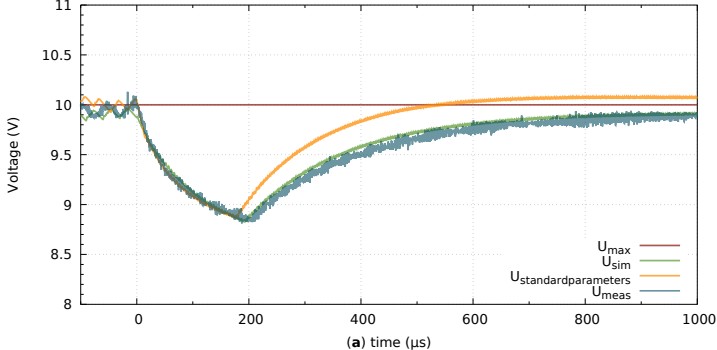

**Figure 12.** *Cont.*

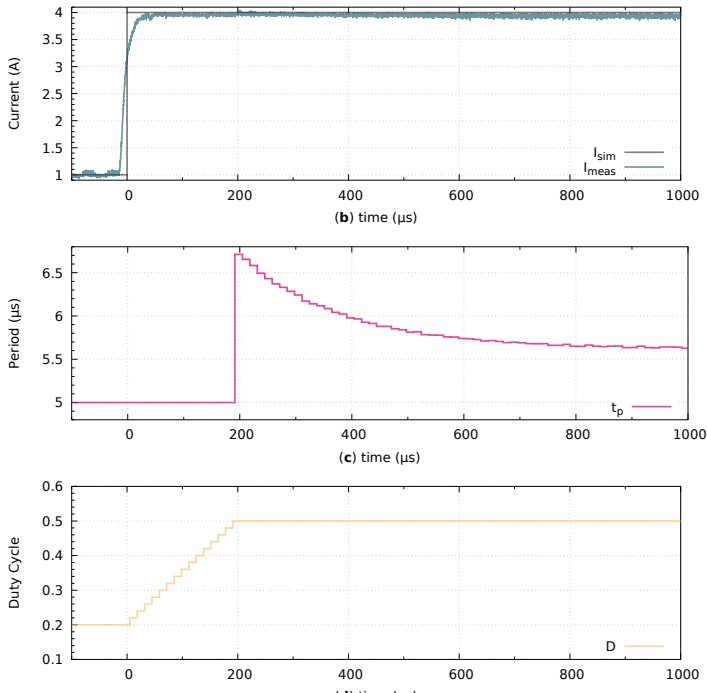

**Figure 12.** The load response of the power supply was measured: the external output current $I_{\text{out}}$ was increased from 1 A to 4 A (**b**) at $t = 0$. The output voltage is measured in (**a**), where an output voltage drop is observed during duty cycle adjustment. The simulated and measured output currents are shown in (**b**), while the simulated switching period $t_p$ is shown in (**c**), and the simulated duty cycle $D$ is shown in (**d**).

## 7.5. CCCV Transition Step Response

In Figure 13, the constant current to constant voltage transition is measured. The effective output capacitance was determined to $C_{\text{out}} = 45\,\mu\text{F}$. The converter was first operated in constant current (CC) mode, and then the transition to constant voltage (CV) was demonstrated. The CCCV control was implemented by choosing the minimal value of two parallel controllers, as seen in Figure 4.

At $t < 0$, the converter operated in CC mode as the output current was limited to 2 A, while the slave controller operated in duty cycle modulation. The load $R_{\text{load}} = 10\,\Omega$ resulted in an output voltage of $U_{\text{out}} = 20$ V. The voltage limit was set to $U_{\text{max}} = 24$ V. At $t = 0$, the current limit $I_{\text{max}}$ was increased from 2 A to 3 A. The master controller demanded additional output current, therefore the duty cycle was increased step wise during duty cycle ramp-up. This prevents false overcurrent triggering. Frequency modulation was used since $t \approx 170\,\mu\text{s}$, when the duty cycle ramp-up was completed, allowing a significantly increased output current and resulting in a faster output voltage rise. When the converter was about to reach its output voltage limit, the switching period was reduced. The converter continued to operate in frequency modulation. The output voltage rose from 20 V within 400 μs to the output voltage limit of $U_{\text{max}} = 24$ V. No output voltage overshoot was observed. A match between simulation and measurement is shown.

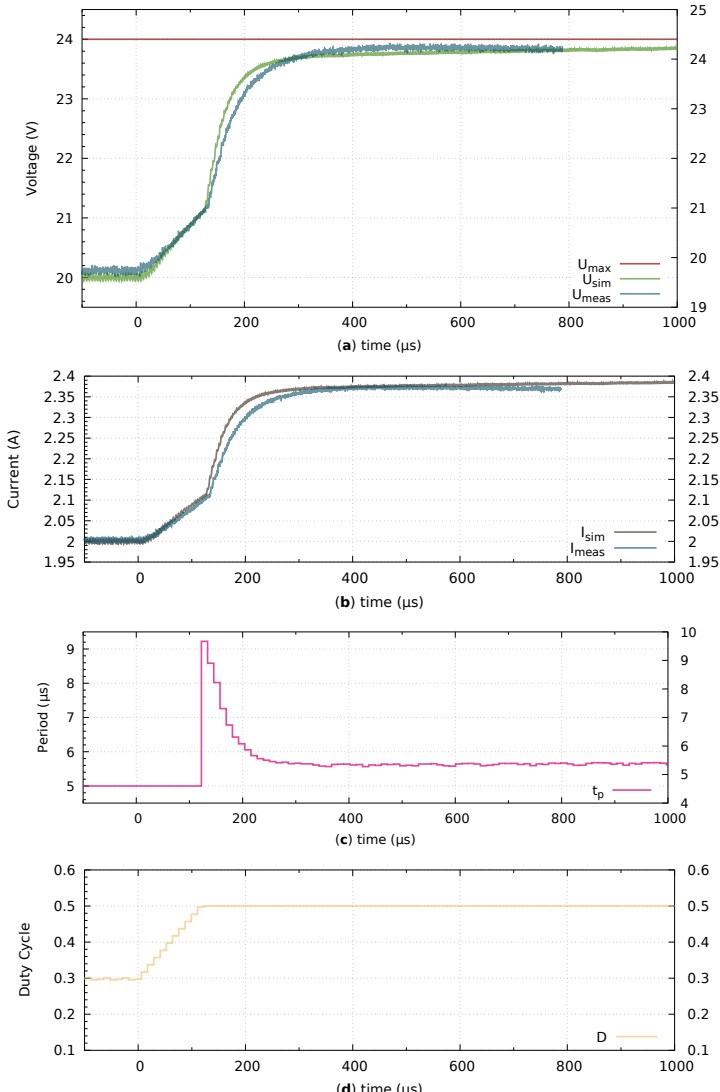

**Figure 13.** The constant current constant voltage operation of the control was verified: the maximal output current $I_{max}$ was increased at $t = 0$ in a step response from 2 A to 3 A in (**b**) to demonstrate the CCCV behavior. The output voltage (**a**) limit was held constant at 24 V. Furthermore, the measured and simulated output voltage are shown. The simulated switching period $t_p$ is shown in (**c**), while the simulated duty cycle $D$ is shown in (**d**).

## 7.6. AC Input Voltage Range

To simulate the AC ripple rejection on the DC link, the converter was extended by a full bridge rectifier and a DC link capacitor $C_{in} = 30\,\mu F$ was used. The AC input voltage had a frequency of 50 Hz at an input voltage of $U_{ac,rms} = 230$ V. The output voltage was set to 25 V. The converter was loaded with a resistor of $10\,\Omega$. The voltage gain was chosen to $K_{pu} = 3$. The output voltage and input voltage are depicted in Figure 14 over half a typical line period. It can be seen that an input voltage range from 270 V to 325 V was accepted. Referring to Figure 14, the maximum switching period limit $t_{p,max}$ was not reached. Hence, a higher ripple is possible.

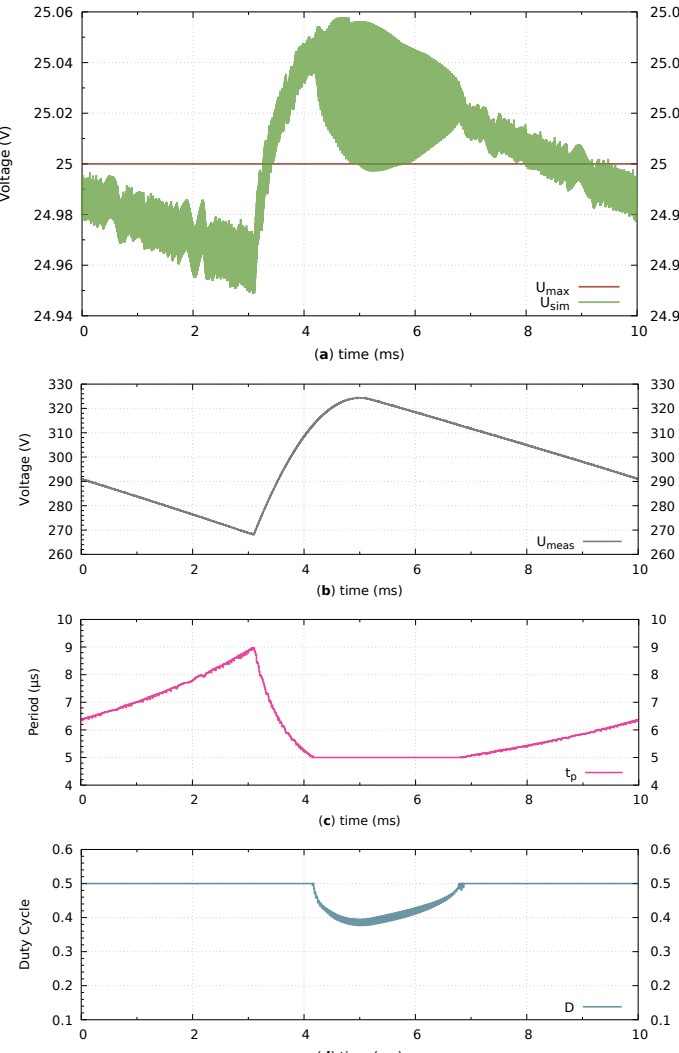

**Figure 14.** The operation of the converter at 50 Hz AC is simulated: The simulated output voltage $U_{\mathrm{sim}}$ is shown in (**a**). The simulated DC link voltage is shown also in (**b**), while the simulated switching period $t_{\mathrm{p}}$ is shown in (**c**), and the simulated duty cycle $D$ is shown in (**d**).

### 7.7. DC Link Ripple Rejection

The input voltage range is extended by switching between the different modulation schemes. The ability to cope with a large voltage ripple allows for a smaller DC link capacitance. This simplifies the construction of electrolytic-free power supplies by replacing these by film capacitors.

The ripple gain is calculated by (15). Its inverse is the so-called ripple attenuation. A ripple gain close to zero results in a higher attenuation $A$. The higher the attenuation $A$, the better the converter rejects the DC link ripple on the output.

$$g = \frac{1}{A} = \frac{\dfrac{U_{\mathrm{output,p2pripple}}}{U_{\mathrm{output,peak}}}}{\dfrac{U_{\mathrm{input,p2pripple}}}{U_{\mathrm{input,peak}}}} \tag{15}$$

The converter ripple gain is measured to $g = 0.02$ and $A = 50$ in Figure 14.

### 7.8. Loop Gain Analysis

The loop gain analysis was conducted in simulation at an output voltage of 24 V and an amplitude of 0.1 V in CV mode. In Figure 15, a bandwidth of 1 kHz is simulated in the low acoustic noise design

using a gain of 1/9. In a loop gain optimized design, using a factor of 1/4, a bandwidth of 3 kHz was observed. The acoustic noise of the loop gain optimized design is seen as jitter in the magnitude.

The converter's open-loop gain in Figure 15 has a typical integrator characteristic and suggests the well-tempered operation of the converter.

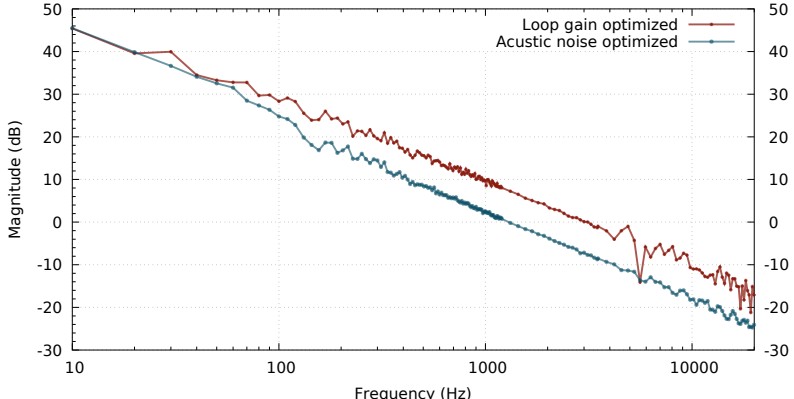

**Figure 15.** The simulated open-loop gain magnitude over frequency is shown for two design choices. The worst case bandwidth frequency is larger than 1 kHz.

## 8. Comparison with the State-of-the-art

In this section, the prototype is compared against the state-of-the-art converters.

### *8.1. Output Voltage Range*

In comparison to conventional voltage-to-voltage converters, a larger output voltage range was achieved. The experiments and simulations demonstrated a large output voltage range from 5 V up to 25 V. This equals an output voltage range of 1:5. Currently, an output voltage range of 1:2 is considered large [9]. Hence, the presented output voltage range is 2.5 times larger compared to state-of-the-art soft-switching converters.

### *8.2. Input Voltage Range*

The converter's input voltage range is shown in Figure 7. Experiments demonstrated an input voltage range from $U_{in,min} = 270$ V to $U_{in,max} = 325$ V. The output current of the converter is increased by choosing a larger resonance capacitor $C_1$ or by lowering the transformer ratio. Hence, a wide input voltage range is possible at the cost of a lower efficiency. Therefore, the SLC topology could be adopted for a large input voltage range.

### *8.3. DC Link Ripple Attenuation*

The proposed converter shows, in Figure 14, a ripple gain of 0.02. Standard LLC converters have a ripple gain of 0.39 [12]. Hence, the presented control attenuates the DC link ripple approximately twenty times better compared to existing solutions.

### *8.4. Control Bandwidth*

Conventional LLC converters have a typical bandwidth of 1.5 kHz [9] to 2 kHz [13] (p. 14) when connected to a restive load. Thus, referring to Figure 15, the presented control has a similar bandwidth compared to a typical LLC converter.

### *8.5. Overshoot*

The cascaded current mode control changes the transient response characteristic of the converter from a state-of-the-art PT2 (second-order lag element) [14] to PT1 (first-order lag element). As one pole

is eliminated in current mode control [6], no overshoot is observed in the prototype. State-of-the-art overshoot optimized designs show an typical overshoot of $\approx 5\%$ [14].

*8.6. Large Signal Step Response*

The response time for CV, CC, and CCCV are very fast with $t_{\text{resp,95\%}} < 400$ µs. State-of-the-art converters feature a load responses time of 150 ms at an output voltage change from 40 V to 50 V [14].

## 9. Conclusions

The paper demonstrates fast and accurate control of a series LC converter in constant current, constant voltage mode and reliable transitions between those operational modes. Ninety-five percent of the output current and voltage target is reached in less than 400 µs during the step response test. The converter attenuates AC ripple on the DC link by a ratio of 1:50. The high attenuation is achieved due to the transfer function (1), canceling out input and output voltage variations. The high attenuation allows for a high DC link ripple. By this, high-capacitance electrolytic capacitors can be replaced with low-capacitance film capacitors without significantly increasing the converter size. Thereby, a significant increase in lifetime is expected.

In contrast to a resonant converter controlled in voltage mode, no overshoot was measured. This is achieved by splitting the controller into a master and a slave controller. The prototype demonstrated an output voltage range from 5 V to 25 V, which equals an extremely large dynamic range from 1:5.

The CCCV characteristic allows for the use of the converter for demanding applications, e.g., laboratory power supplies or lithium battery chargers. The CCCV control is implemented using a cascaded control loop: the master controller sets the SLC current and the open-loop slave controller controls the switching period, duty cycle, and pulse skipping. The converter is stable over the whole operation range, from no load to heavy load conditions.

## 10. Patents

The modulation scheme, which is based on Equation (1) is covered by a pending patent. Germany: DE 10 2018 216 749.4—"Verfahren zur Steuerung eines Serien-Resonanz-Wandlers".

**Author Contributions:** Conceptualization, M.H.; methodology, M.H.; simulation, M.H.; validation, Q.X.; formal analysis, W.H.; investigation, Q.X., M.H.; writing, original draft preparation, M.H.; writing, review and editing, C.S., F.D., S.E., W.H.; visualization, M.H.; supervision, R.K.; project administration, R.K.

**Funding:** This research received no external funding.

**Conflicts of Interest:** The authors declare no conflict of interest.

## Abbreviations

The following abbreviations are used in this manuscript:

| | |
|---|---|
| CC | Constant current |
| CCCV | Constant current, constant voltage |
| CV | Constant voltage |
| DSP | Digital signal processor |
| MLCC | Multilayer ceramic capacitor |
| PT1 | First-order lag element |
| PT2 | Second-order lag element |
| PWM | Pulse width modulation |
| SMPS | Switch mode power supply |
| SL | Series LC (inductor capacitor) |
| SLCC | Series LC (inductor capacitor) converter |

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
