# Peer review of "Current Mode Control of a Series LC Converter Supporting Constant Current, Constant Voltage (CCCV)"

_energies, doi:10.3390/en12142793_

Reviewer 1 Report

A state of the art must be included.

Experimental results must be compared with previous controllers.

Author Response

Dear reviewer, thank you for your feedback.

1) Include state of the art:

A section "state of the art" was included.

2) Comparison with existing controllers:

A corresponding section is added.

Reviewer 2 Report

- Table I should show the output power of the tests.

- English should be improved as there are some sentences that can be improved.

- Fig 7 should have something to know the dimensions, some ruler or pen to get the scale.

- In Fig. 5, what is the input to the first box Limit Imax?

Author Response

Dear reviewer, thank you for your feedback.

1) Output power of the test.

The output power is highly varying with different output voltages. Hence, it's not a fixed value. Therefore, we added only the maximum output power to Table 1.

2) Improve English:

We read again our paper and made some minor (non-highlighted) changes regarding the language. In case we missed some points, please point us to it and help us with your language skills.

3) Size of prototype:

We added the PCB dimensions to the captions of the prototype. I tried a photo with a ruler. However, a ruler onscreen disturbs the quality of the image.

4) In Fig. 5, what is the input to the first box Limit Imax?

Thanks for pointing us to this non-clearity. We fixed Figure 5 and the description.

Reviewer 3 Report

This paper aims to introduce a new control algorithm for soft-switching series LC converters. The controller is split into a master and a slave controller, where the master implements constant-current-constant-voltage (CCCV) control whereas the slave controller implements the open-loop current control generating the PWM parameters.

1) It is claimed that in contrast to typical soft-switching power supplies a large input and output voltage range can be achieved. However the section about results lacks a comparison against the state of the art solutions and a clear discussion explaining why the proposed controller outperforms the state of the art.

2) Deviations between simulated and measured curves is in some cases rather significant. For example in Fig. 10 the deviation should be better explained.

3) The commutation between the different control regions, namely: frequency modulation, duty cycle modulation and pulse skipping should be better explained in the experiments shown in the sections about results, in particular in section 5.4 it is not clearly explained how the different controls are excited. 

4) It is shown that the converter operating in pulse skipping mode generates significant output ripple. Can the voltage ripple be reduced by using a proper filter?

Author Response

Dear reviewer, thank you for your detailed review. Your critical mindset helped me to improve the paper. Several sections have been added based on your comments.

1) We added a section, comparing the presented technology with state of the art research.

2) I investigated Figure 10 again, and found inaccurate parameters. Now the simulation matches fairly well, as all the other simulations.

3) The commutation between the individual control regions is explained in detail now.
It's discussed in the slave controller section and the individual sections of the corresponding tests.

4) Output Filter: You mention an interesting topic for further investigation. It should be identified, how far an output filter would affect the corresponding control loop. I mentioned this in the paper.

Round  2

Reviewer 1 Report

Thank you for addressing my previous comments. The article has been improved.

Reviewer 3 Report

In the revised version of the paper the authors addressed my comments in a very satisfactory way. The revised version is therefore recommended for publication.